



# Silicone tube humidity generator

Robert F. Berg[1], Nicola Chiodo[2], Eric Georgin[3]

[1] NIST, 100 Bureau Drive, Gaithersburg, Maryland 20899 USA
[2] LCM LNE-CNAM, Laboratoire Commun de Métrologie LNE- CNAM, La Plaine-Saint Denis F93210, France
[3] CETIAT, Centre Technique des Industries Aerauliques et Thermiques, Villeurbanne, France

*Correspondence to*: Robert F. Berg (robert.berg@nist.gov)

**Abstract.** We describe the model and construction of a two-flow (or divided-flow) humidity generator, developed at LNE-CNAM, that uses mass flow controllers to mix a stream of dry gas with a stream of humid gas saturated at 28 °C. It can generate
a wide range of humidity, with mole fractions in the range $0.7 \times 10^{-6} < x < 9000 \times 10^{-6}$, without using low temperature or high pressure. This range is suitable for calibrating balloon-borne instruments that measure humidity in the stratosphere, where $x \sim 5 \times 10^{-6}$. The generator's novel feature is a saturator that comprises 5 m of silicone tubing immersed in water. Water enters the humid gas stream by diffusing through the wall of the tubing until the gas stream flowing through the tubing is saturated. This design provides a simple, low-cost humidity generator with an accuracy that is acceptable for many applications. The key
requirement is that the tubing be long enough to ensure saturation, so that the saturator's output is independent of the dimensions and permeability of the tube. A length of only a few meters was sufficient because the tube was made of silicone; other common polymers have permeabilities that are 1000 times smaller. We verified the model of the transition from unsaturated flow to saturated flow by measuring the humidity while using three tube lengths, two of which were too short for saturation. As a more complete test, we used the generator as a primary device after correcting the calibrations of the mass
flow controllers that determined the mixing ratio. At mole fractions $50 \times 10^{-6} < x < 5000 \times 10^{-6}$, the generator's output mole fraction $x_{gen}$ agreed to within 1 % with the value $x_{cm}$ measured by a calibrated chilled-mirror hygrometer; in other words, their ratio fell in the range $x_{gen}/x_{cm} = 1.00 \pm 0.01$. At smaller mole fractions, their differences fell in range $x_{gen} - x_{cm} = \pm 1 \times 10^{-6}$.

# 1 Introduction

A hygrometer can be calibrated by comparing it to one that either is a primary standard or has already been calibrated. Alternatively, it can be compared to a primary humidity generator. All these comparisons require a steady flow of humid gas. This can be a challenge when the humidities of interest range from saturation, which at room temperature is a mole fraction of $x \approx 0.03$, down to less than $x \approx 1$ ppm for gases used by the semiconductor industry. (1 ppm is a mole fraction of $10^{-6}$.)






There are three types of primary humidity generators:

- A single-pressure generator saturates the gas at pressure $P$ and temperature $T_1$ below room temperature and then outputs the gas at a higher temperature $T_2$, typically near room temperature. The resulting water mole fraction is approximately $x = P_V(T_1)/P$, where $P_V(T_1)$ is the vapor pressure of water at temperature $T_1$. See, for example, Meyer (2008), Scace (2001), Wettstein (2018), and Cuccaro (2018).

- A two-pressure generator saturates the gas at some high pressure $P_1$ at temperature $T_1$ and then outputs the gas at a lower pressure $P_2$, often 1 atmosphere. The resulting water mole fraction is approximately $x = P_V(T_1)\,(P_2/P_1)$. The pressure ratio is limited to typically $P_2/P_1 > 0.1$, so generating a low humidity requires also a low-temperature saturator. See, for example, Wexler (1952), Hasegawa (1977), and Meyer (2008).

- A two-flow (or divided-flow) generator mixes a known flow rate $\dot{n}_{dry}$ of a dry gas with either a known flow rate of liquid water (Vega-Maza, 2012) or, more frequently, a known flow rate $\dot{n}_{wet}$ of a gas of known water mole fraction $x_{wet}$. When $\dot{n}_{wet} \ll \dot{n}_{dry}$, the resulting mole fraction is approximately $x = x_{wet}(\dot{n}_{wet}/\dot{n}_{dry})$. Table 1 gives examples of primary two-flow humidity generators.

Table 1. Examples of primary two-flow humidity generators. Except for (Wexler, 1952), the two flows were controlled by mass flow controllers (MFC).

| reference | flow divider | flow ratio minimum | flow ratio uncertainty |
|---|---|---|---|
| Wexler 1952 | 6 orifices | 0.17 | 0.03 |
| Takahashi 1996 | 1 wet + 1 dry + 1 total MFC | 0.05 | 0.008 |
| Weremczuk 2008 | 4 wet + 1 dry MFC | 0.00001 | not stated |
| Meyer 2008 | 7 wet + 1 dry MFC | 0.000002 | 0.0006 |

The device described here is a primary two-flow humidity generator. This type has the advantage that it can generate a wide range of humidity without using low temperature or high pressure. By contrast, to achieve $x = 1$ ppm, the saturator of a single-pressure generator must operate at -76 °C. A two-pressure generator cannot achieve such a low humidity unless it also uses a cold saturator.

The present generator has a novel saturator that comprises 5 m of silicone tubing immersed in water. Water enters the humid gas stream by diffusing through the wall of the tubing until the gas stream flowing through the tubing is saturated. This design provides a simple, low-cost humidity generator with an accuracy that is acceptable for many applications. The key requirement is that the tubing be long enough to ensure saturation, so that the saturator's output is independent of the dimensions and permeability of the tube. A length of only a few meters was sufficient because the tube was made of silicone (polydimethylsiloxane or PDMS); other common polymers have permeabilities that are 1000 times smaller. See the appendix.





We built the generator at LNE-CNAM to supply a flow of humid gas that would allow us to compare an experimental microwave hygrometer (Merlone, 2017) to a calibrated hygrometer over our range of interest, 1 ppm $< x <$ 10000 ppm. Our immediate need was a source of humid gas that was stable, compact, and inexpensive. After meeting that need, we realized that the generator could function also as a primary source whose uncertainty would be limited by the uncertainty of the humid/dry flow ratio. The saturator used in the present generator is much less expensive than those used in previous generators

because it is simply constructed from inexpensive commercial components. Its small size also facilitates its temperature control, which in principle could be as simple as a small ice bath at 0 °C.

Our model of the generator assumed that the gas exiting the silicone tube was completely saturated. We tested that assumption by measuring the humidity with three tube lengths, two of which were too short for saturation. As a more complete test, we

demonstrated that, when the generator was used as a primary device, it agreed with the calibrated hygrometer to within the uncertainty of the mass flow controllers that determined the mixing ratio.

In the following, we first describe the model and construction of the generator and how it can be used as a primary standard. After demonstrating that the degree of saturation depends on the length of the silicon tube, we compare the generator's

performance to the calibrated hygrometer. That comparison required two small corrections of the model due to pressure drops in capillaries and the diffusion of water through the carrier gas. An additional small correction was peculiar to the saturator: the diffusion of the carrier gas through the wall of the silicone tubing. The last section describes the generator's uncertainty.

## 2 Model of the humidity generator

### 2.1 Permeation of water through silicone

For each small length $dz$ of the tube, permeation causes water to flow through the tube wall at the following molar flow rate (Crank, 1975):

(1)  $$d\dot{n}_{\text{perm}}(z) = \frac{2\pi\varphi}{\ln(d_{\text{out}}/d_{\text{in}})}[P_V(T) - p_{\text{H2O}}(z)]dz .$$

Here, $\varphi$ is the $H_2O$ permeability of the tube material, $d_{\text{out}}$ and $d_{\text{in}}$ are the tube's outer and inner diameters, $P_v(T)$ is the vapor pressure of water at the temperature $T$ of the saturator, and $p_{\text{H2O}}(z)$ is the partial pressure of water at position $z$ inside the tube.

(The notation uses $p$ for partial pressure and $P$ for total pressure.) At 25 °C the vapor pressure of water is $P_v = 3.2$ kPa. Equation (1) assumes that the temperature and total pressure are the same inside and outside the tube, so that the permeation flow is driven by only the partial pressure of water. The permeation flow is zero when the partial pressures inside and outside the tube are equal, *i.e.*, when the chemical potentials are in equilibrium.



At the tube entrance ($z = 0$) the gas is dry, and at the tube exit ($z = L$) the gas is humid. As explained later, the actual mole fraction at the entrance was 0.43 ppm. Solving Eq. (1) gives the water partial pressure at the exit:

(2) $$p_{\text{wet}} = P_V \left[ 1 - \exp \left( -\frac{L}{L_{\text{sat}}} \right) \right].$$

Here $L_{\text{sat}}$ is the saturation length given by

(3) $$L_{\text{sat}} = \frac{\ln(d_{\text{out}}/d_{\text{in}})}{2\pi RT\varphi} \dot{V}_{\text{wet}},$$

where $R$ is the universal gas constant, and $\dot{V}_{\text{wet}}$ is the volume flow rate of the gas. For a long tube with $L \gg L_{\text{sat}}$, the exiting gas is saturated and the water partial pressure equals the vapor pressure. For clarity, we relate the volume flow rate $\dot{V}$ to the molar flow rate $\dot{n}$ by defining the volume flow rate to be that of an ideal gas, namely

(4) $$\dot{V} = \frac{RT}{P} \dot{n}.$$

A mass flow controller (MFC) often reports the molar flow rate as a volume flow rate by using a reference temperature and

pressure, say $T_0 \equiv 273.15$ K and $P_0 \equiv 101325$ Pa.

As shown in Figure 1, mixing the humid gas flow with a dry gas flow produces the total flow rate, $\dot{V}_{\text{total}} = \dot{V}_{\text{wet}} + \dot{V}_{\text{dry}}$. If the total pressure after mixing is $P$, then the final mole fraction is:

(5) $$x = \frac{p_{\text{wet}}}{P} \frac{\dot{V}_{\text{wet}}}{\dot{V}_{\text{total}}} = \frac{P_V}{P} \frac{\dot{V}_{\text{wet}}}{\dot{V}_{\text{total}}} \left[ 1 - \exp \left( -\frac{L}{L_{\text{sat}}} \right) \right],$$

To understand Eq. (5), consider its two limits. If the permeability is small, then $L \ll L_{\text{sat}}$, and

(6) $$x \cong \frac{2\pi\varphi L P_V}{\ln(d_{\text{out}}/d_{\text{in}})\dot{n}_{\text{total}}}.$$

In this limit, the mole fraction $x_{\text{H2O}}$ depends on the tube dimensions and on temperature through the factors $P_v(T)$ and $\varphi(T)$. Some commercial low-humidity reference standards operate in this limit (Permeation tube generators, disclaimer).

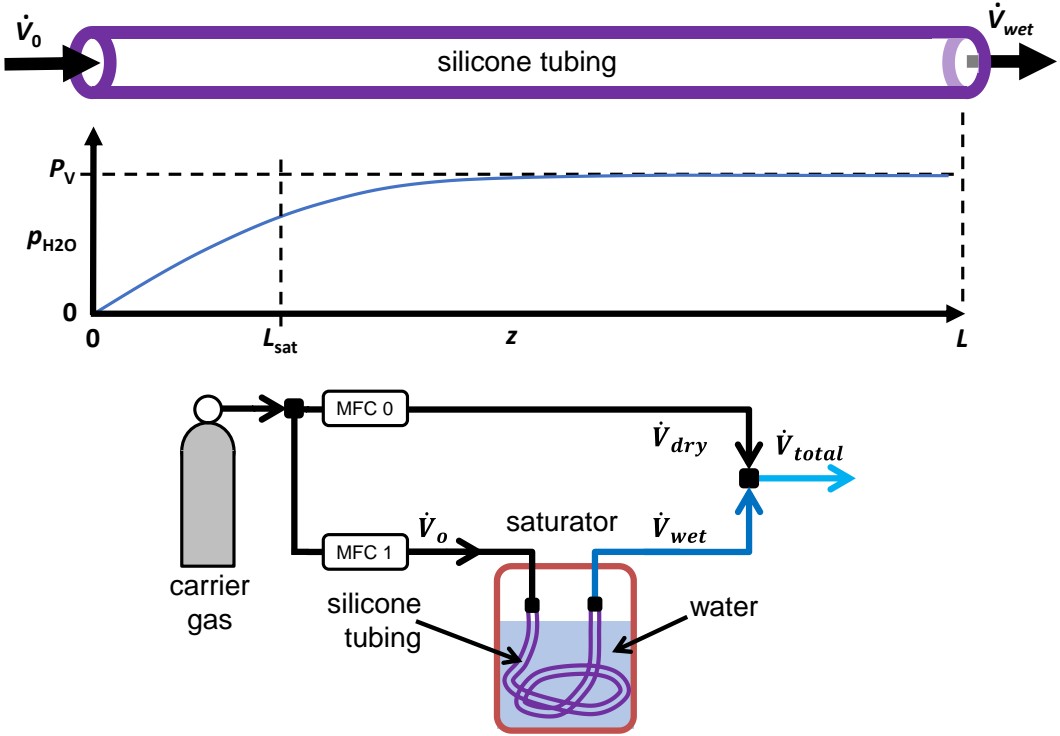

Figure 1. UPPER: Dry gas that enters a permeable tube immersed in water becomes more humid. The exiting gas is saturated if the tube is much longer than the characteristic length $L_{\text{sat}}$. LOWER: The humidity generator combines a stream of humid gas with a stream of dry gas.

## 2.2 Using the generator as a primary standard

The opposite limit of Eq. (5), $L \gg L_{\text{sat}}$ , occurs when the permeability is large, so that

$$(7) \qquad x = \frac{P_V}{P} \frac{\dot{n}_{\text{wet}}}{\dot{n}_{\text{total}}} .$$

Operating in this limit has two advantages. First, it is simple. The mole fraction depends on the two ratios, $P_V/P$ and $\dot{n}_{\text{wet}}/\dot{n}_{\text{total}}$, and not on the material properties or dimensions of the tube. Second, it offers the possibility of using the generator as a primary standard because the vapor pressure $P_v(T)$ is well known, the total pressure $P$ can be measured accurately, and a ratio of flow rates can sometimes be obtained with an uncertainty that is smaller than the uncertainties of the individual flow meters. The smaller uncertainty of the ratio is possible if the individual uncertainties account for a possible scale error that is common to both flow meters. For example, if the standard used to calibrate both flow meters was in error by 2 %, that error will cancel out of the ratio.

Equation (5) assumes that one knows the flow $\dot{V}_{\text{wet}}$ at the generator exit, but in practice one knows only the flow $\dot{V}_0$ at the generator entrance. The generator adds a water vapor flow $\dot{V}_{\text{H2O}}$, so that, in the ideal-gas limit, the flow exiting the tube is





(8) $$\dot{V}_{\text{wet}} = \dot{V}_0 + \dot{V}_{\text{H2O}} = \frac{\dot{V}_0}{1-p_{\text{wet}}/P} .$$

Using Eq. (2) in Eq. (8) gives the total flow rate exiting the generator:

(9) $$\dot{n}_{\text{total}} = \dot{n}_{\text{dry}} + \dot{n}_0 \left[ 1 - \frac{P_V}{P} \left( 1 - \exp\left( -\frac{L}{L_{\text{sat}}} \right) \right) \right]^{-1} .$$

Using Eq. (9) in Eq. (5) gives the mole fraction

(10) $$x = \frac{\frac{P_V}{P}\frac{\dot{n}_0}{\dot{n}_{\text{dry}}}}{1 + \frac{\dot{n}_0}{\dot{n}_{\text{dry}}} - \frac{P_V}{P}\left[ 1 - \exp\left( -\frac{L}{L_{\text{sat}}} \right) \right]} ,$$

which in the limit of saturation is

(11) $$x = \frac{\frac{P_V}{P}\frac{\dot{n}_0}{\dot{n}_{\text{dry}}}}{1 + \frac{\dot{n}_0}{\dot{n}_{\text{dry}}} - \frac{P_V}{P}} .$$

This expression depends only on the ratio of the vapor pressure of water $P_v$ to the measured total pressure $P$, and the ratio of
flow rates $\dot{n}_0/\dot{n}_{\text{dry}}$.

## 2.3 The enhancement factor

The preceding equations assume ideal-gas behavior and do not account for four effects: (1) the nonideal behavior of the carrier
gas and water vapor, (2) intermolecular forces in the mixture of water vapor and carrier gas, (3) the action of the pressure on
the condensed water (the Poynting effect), and (4) the solution of the carrier gas in the water (the Raoult effect). (See, for
example, Hyland (1975), Wylie (1996), Koglbauer (2007), Koglbauer (2008), Lovell-Smith (2016).) Correcting Eq. (11) for
these effects can be accomplished by the change

(12) $$\frac{P_V}{P} \longrightarrow f_W \frac{P_V}{P} .$$

where $f_w$ is the "pressure enhancement factor". With this correction, Eq. (11) becomes

(13) $$x = \frac{f_W\frac{P_V}{P}\frac{\dot{n}_0}{\dot{n}_{\text{dry}}}}{1 + \frac{\dot{n}_0}{\dot{n}_{\text{dry}}} - f_W\frac{P_V}{P}} .$$

The carrier gas in the present study was argon, the same gas that was used in the Boltzmann project to redefine the kelvin
(Pitre, 2017). The spherical resonators used in that project are similar to those used in the differential microwave hygrometer
shown in **Figure 2**. Unlike for air, the values of $f_w$ for argon have not been measured directly, but there are measurements near
25 °C (Koglbauer, 2008) of a related quantity, the concentration enhancement factor $g_w$. Those measurements agree with values
of $f_w$ that we obtained from calculated values of the argon-water virial coefficient $B_{aw}$. In addition to being available over a
wider range of temperature, the calculated values have a smaller uncertainty than the measured values. We will not discuss





further the values of $f_w$ for argon, but we note that the present corrections due to the enhancement factor were less than 1.0 %, and they contributed less than 0.3 % to the relative uncertainty of the humidity.

## 3 Apparatus

The experimental setup, shown in Figure 2, comprised a thermal enclosure, the humidity generator, a commercial chilled-
mirror hygrometer, the gas manifold, and the electronic instrumentation. The humidity generator was held in the thermal enclosure.

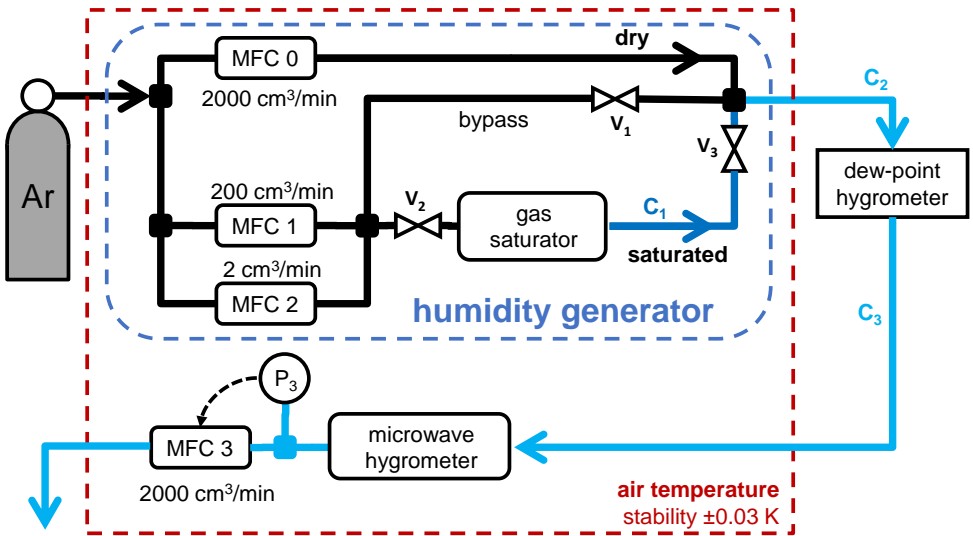

Figure 2. The experimental setup. MFC = mass flow controller. V = bellow-sealed valve. C = capillary. The output of the
humidity generator went to the commercial chilled-mirror hygrometer and then to MFC 3, which controlled the exit pressure at $P_3 = 108$ kPa. (The microwave hygrometer (Merlone, 2017), was not used.)

### 3.1 Gas manifold

Gas from the Ar bottle ($x < 0.5$ ppm from specifications) entered the thermal enclosure and the humidity generator. The output of the humidity generator exited the thermal enclosure and went to the chilled-mirror hygrometer through a 1 m capillary with
an inner diameter of 1.3 mm. The output of the hygrometer re-entered the thermal enclosure through a similar capillary. The MFC at the exit (maximum flow 2000 cm$^3$ min$^{-1}$) was controlled by an algorithm that stabilised the pressure $P_3$ measured just before the MFC by a precise pressure gauge (Paroscientific 745-100A).

We minimised the number of components to avoid unwanted sources of water vapor; for example, we removed an over-
pressure safety valve because atmospheric water vapor could penetrate its rubber seal. We used small-diameter capillaries to



limit the surface area for adsorption and desorption. Even after reducing the surface area of the stainless-steel components, the time to achieve a humidity below 10 ppm was several hours.

## 3.2 Thermal enclosure

The thermal enclosure, which was based on the description given in (Berg,2014), controlled the temperatures of the humidity generator, the gas manifold, the MFCs, and the pressure gauges near 28 °C. Its height and area were respectively 0.63 m and $(1.27 \times 0.66)$ m$^2$, and its walls were rigid sheets of 25 mm polyisocyanurate insulation. The temperature was stabilized by a commercial controller (Arroyo Instruments model 5305 (disclaimer)) that drove a thermoelectric cooler (Laird model AA-034-12-22). The control thermistor was located in the stream of air that entered the cooler. Measurements at other locations found that the temperature was stable throughout the enclosure to approximately ±0.03 K. Without the cooler, the power dissipated by the enclosed instruments would raise the temperature to 6 K above ambient. Despite the use of four small stirring fans, the enclosed electronics created a gradient of 1 K in the enclosure, with the coldest point located at the output of the thermoelectric cooler.

The thermal enclosure was made large enough to include the temperature-sensitive microwave hygrometer and its associated tubing. A much smaller enclosure containing only the saturator would have been sufficient to keep its output constant.

## 3.3 Humidity generator

The humidity generator comprised the saturator and three MFCs. It first divided the input gas into two streams, dry and saturated. The dry stream was controlled by an MFC with a maximum flow of 2000 sccm (Bronkhorst F-201CV). (1 sccm $\cong$ 0.74 µmol s$^{-1}$ is the molar flow rate corresponding to 1 cm$^3$ min$^{-1}$ of an ideal gas at 0 °C and 1 atm.) The saturated stream was controlled by two MFCs in parallel, one with a maximum flow of 2 cm$^3$ min$^{-1}$ and the other with 200 cm$^3$ min$^{-1}$ (Alicat MC). The saturated stream mixed with the dry stream after traveling through a capillary heated to 20 K above the enclosure temperature to avoid condensation. The range of possible mole fraction was 0.7 ppm $< x <$ 9000 ppm, where the minimum mole fraction corresponded to the minimum MFC setting of 0.015 cm$^3$ min$^{-1}$. We operated the generator with total flow rates from 50 cm$^3$ min$^{-1}$ to 300 cm$^3$ min$^{-1}$.

**Table 2** gives the specified performances of the MFCs, which we did not verify directly. However, as discussed later, comparing the generator to the dew-point hygrometer showed that the MFC calibrations had drifted outside their specifications. Fortunately, the comparison data had sufficient redundancy that the drifts could be modelled by a linear function of flow rate, so that the generator's performance could still be tested. Recognizing the possibility of MFC errors is important, and the discussion on uncertainty suggests various ways to reduce such errors.



Table 2. Specified standard uncertainty of the mass flow controllers. 1 sccm $\cong$ 0.74 μmol s$^{-1}$ is the molar flow rate corresponding to 1 cm$^3$ min$^{-1}$ of an ideal gas at 0 °C and 1.01325 bar.

|  | function | range [sccm] | specification |
|---|---|---|---|
| MFC 0 | dry gas stream | 2000 | 2 sccm + 0.5 % of reading |
| MFC 1 | saturator input | 200 | 0.2 sccm + 0.4 % of reading |
| MFC 2 | saturator input | 2 | 0.002 sccm + 0.4 % of reading |
| MFC 3 | pressure control | 2000 | 2 sccm + 0.5 % of reading |

### 3.4 Saturator

The saturator consisted of a 1-litre glass dewar that contained a commercial sealed platinum resistance thermometer (pt1000) and 5.07 m of silicone (PDMS) tubing (Saint-Gobain product Versilic), with inner and outer diameters of 4 mm and 6 mm. Although the tubing had an acid-acrylic odour, we assumed that any outgassing had a negligible effect. Both were immersed in chromatography grade or "ultra-pure" water. See Figure 3. The saturator temperature and pressure were not directly controlled. The pressure drop along the tubing was negligible, ~60 Pa for a flow of 200 sccm. The water temperature was typically 0.5 K higher than the surrounding enclosure due to the heat dissipated by nearby components, especially that of the heated capillary. The standard uncertainty of the thermometer's calibration was 0.021 °C. We assumed that stratification of the water temperature contributed negligible additional uncertainty because the thermometer was located near the exit of the tube; this assumption was supported indirectly by the consistency obtained when using different flow rates through the saturator. A simpler way to control the saturator temperature would have been to use an ice bath in the dewar. Three bellows-sealed valves were used to isolate the saturator for various tests.

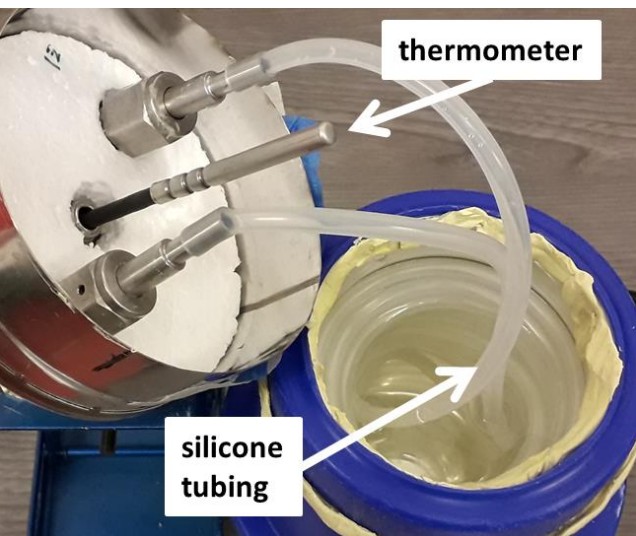

Figure 3. The silicone tubing was loosely coiled in a 1-litre dewar filled with water.





**3.5 Reference hygrometer.**

The humidity generator was compared against a commercial chilled-mirror hygrometer (MBW model 373LX (disclaimer)) with an accuracy of 0.1 °C in the nominal flow range from 200 cm$^3$ min$^{-1}$ to 2000 cm$^3$ min$^{-1}$. The hygrometer determined the water mole fraction from the hygrometer's measurements of pressure and dew-point temperature. The manufacturer's calibration with air spanned dew/frost points from −90 °C to +20 °C; a calibration made 8 months later by CETIAT found

deviations of less than 0.1 K. We adjusted the hygrometer's values by the small difference between the enhancement factors of air and argon:

(14) $\quad f_w(\mathrm{Ar}) - f_w(\mathrm{air}) = -0.0005 + (1.1 \times 10^{-5}\ \mathrm{K}^{-1})(T - 273.15\ \mathrm{K})$ .

**4 Experimental results**

The performance of the generator was evaluated by comparing the mole fraction $x$ expected at its output with the mole fraction $x_{\mathrm{cm}}$ measured by the chilled-mirror hygrometer. The expected mole fraction was calculated from Eq. (10), which depended on the pressure $P$, the temperature $T$ of the saturator, and the flow rates $\dot{V}_0$ and $\dot{V}_{\mathrm{dry}}$. The temperature affected the enhancement factor $f_W$ as well as the vapor pressure $P_V$.

The pressure in the saturator was kept as low as possible to minimize the enhancement factor. It was typically 3 kPa higher than the exit pressure, $P_3 = 108$ kPa, due to the impedances of capillary 2 and capillary 3. Exit pressures below 108 kPa could not drive enough flow through MFC 3. The saturator temperature was typically 28 °C.

**4.1 Demonstration of saturation**

Figure 4 shows how the saturator performance depended on the length $L$ of the silicone tube. The tube was held in water at

28.5 °C, the dry flow rate $\dot{V}_{\mathrm{dry}}$ was fixed at 200 cm$^3$ min$^{-1}$, and the wet flow rate $\dot{V}_{\mathrm{wet}}$ was varied by controlling MFC 1 and MFC 2. The reference hygrometer measured the water mole fraction of the resulting mixture. It was not used to measure the mole fraction of the gas stream exiting the saturator because the temperature of the saturator (28.5 °C) was above that of the hygrometer. Also, adding the dry flow also ensured that flow rate of the mixture fell within the reference hygrometer's nominal flow range (200 cm$^3$ min$^{-1}$ to 2000 cm$^3$ min$^{-1}$).


Three tube lengths were used: 6 cm, 50 cm, and 507 cm. The mole fraction calculated for these lengths agreed with the mole fraction measured by the reference hygrometer when the water permeability $\varphi$ of the tube was set at $(950 \pm 50) \times 10^{-14}$ mol s$^{-1}$ m$^{-1}$ Pa$^{-1}$. This value is consistent with those found elsewhere for PDMS; see **Table** in the appendix. All three tubes agree at low flow rates, where the slope derived from Eqs. (7) and (12) is simply



(15) $$\frac{dx}{d\dot{V}_{\text{wet}}} = \frac{1}{\dot{V}_{\text{total}}} f_W \frac{P_V}{P} .$$

Figure 4 shows that the model agrees with all the measurements. At lower flow rates the mole fraction was the same for all three tube lengths $L$, which means that the partial pressure at the tube exits, $p_{H2O}(L)$, was the same for all three tube lengths. Equation (1) assumes that the values of $T$ and $P$ at the exit were equilibrated across the tube wall, in which case the common partial pressure was simply the vapor pressure of water. In other words, $p_{H2O}(L) = P_V$, and the gas at the tube exit was saturated.


Figure 4 demonstrates that the tube length of 507 cm used for the measurements described below was sufficient to saturate the gas. According to Eq. (3), a gas flow as large as 10 sccm, which produced a mole fraction of $x \approx 2000$ ppm, caused a saturation length, the $1/e$ length of Eq. (2), of only 45 cm.

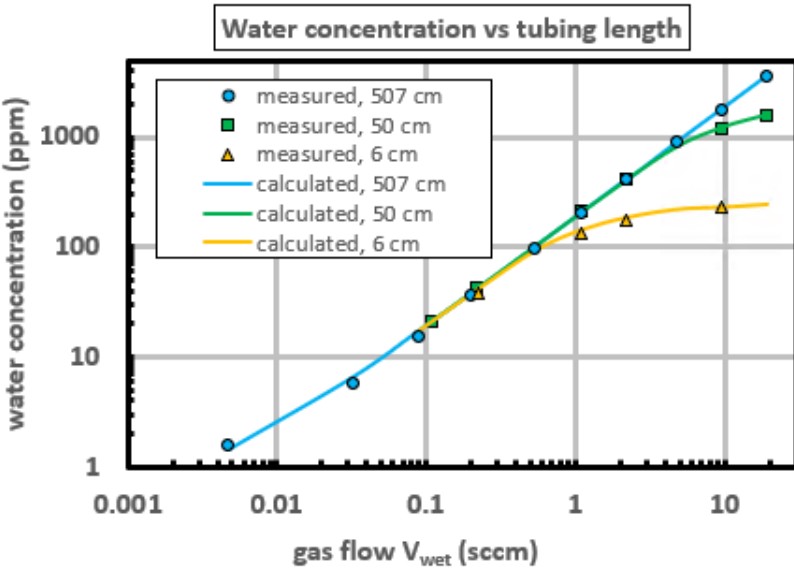


Figure 4. The water mole fraction as a function of the wet flow rate $\dot{V}_{\text{wet}}$ leaving the saturator. The PDMS tube had an inner diameter of 4 mm, an outer diameter of 6 mm, and various lengths. The tube was in water at 28.5 °C, and the dry flow rate was fixed at 200 sccm. (The mole fraction of the lowest point was increased slightly by the mole fraction of the "dry" stream, $x_{dry} = 0.43$ ppm.)

**4.2 Operation as a primary humidity generator**

These tests were performed with the thermal enclosure temperature controlled at 28.0 °C, corresponding to water temperatures in the uncontrolled dewar in the range $(28.8 \pm 0.1)$ °C. The humidity was varied by changing the ratio $\dot{V}_0/\dot{V}_{\text{dry}}$ while keeping constant the sum $\dot{V}_0 + \dot{V}_{\text{dry}}$. The total flow rate was varied from 50 cm$^3$ min$^{-1}$ to 300 cm$^3$ min$^{-1}$, and the saturator input was varied from 0.015 cm$^3$ min$^{-1}$ to 50 cm$^3$ min$^{-1}$ at the points given in Table 3. To ensure equilibrium, each point was an average

taken during 24 hours.





Table 3. Flow rates used for operation as a primary humidity generator.

| total flow rate $\dot{V}_0 + \dot{V}_{dry}$ | saturator input $\dot{V}_0$ MFC 1 | saturator input $\dot{V}_0$ MFC 2 |
|---|---|---|
| (cm$^3$ min$^{-1}$) | (cm$^3$ min$^{-1}$) | (cm$^3$ min$^{-1}$) |
| 50 | 0.1, 0.2, 0.5, 1, 2 | |
| 100 | 0.1, 0.2, 0.5, 1, 2 | |
| 200 | 0.015, 0.025, 0.05 | |
| 200 | 0.1, 0.2, 0.5, 1, 2 | |
| 200 | | 1, 2, 5, 10, 20, 50 |
| 300 | 0.1, 0.2, 0.5, 1, 2 | |

For each pair of wet and dry flow rates, Eq. (13) was used to calculate the mole fraction $x$ from the temperature $T$ and pressure $P$ of the saturator and the ratio of flow rates $\dot{n}_0/\dot{n}_{dry}$. The calculated values were obtained by making four small corrections to the model, as discussed in the following section.

### 4.3 Four corrections to the model

The model, Eq. (13), required corrections for the humidity of the input "dry" gas, the permeation of argon through silicone, the diffusion of water vapor through argon, and the pressure drops caused by flow through capillaries. The first correction was simply

$$(16) \qquad x \rightarrow x + x_{dry} \, ,$$

where $x_{dry} = 0.43$ ppm was the mole fraction measured with zero gas output from the saturator (valve V$_3$ closed).

The second correction seemed necessary because, upon opening the dewar, we always observed gas bubbles on the external surface of the immersed tubing. We attributed that effect to permeation of the carrier gas through the walls of the tubing. The argon permeation may have caused a second effect: small temperature spikes in the saturator temperature that disappeared when the tubing was removed. Perhaps they occurred when a rising bubble changed the temperature distribution in the water. In any case, the spikes had a negligible effect on the average temperature because they had amplitudes < 0.1 K and they occurred only a few times per hour.

As discussed in the appendix, the permeability $\varphi_{Ar}$ of argon through PDMS allowed the carrier gas to diffuse out of the tube and into the surrounding water at the rate

$$(17) \qquad \dot{n}_{perm,Ar} = \frac{2\pi L(p_{out}-p_{in})}{\ln(d_{out}/d_{in})} \varphi_{Ar} \, ,$$

where $p_{in}$ and $p_{out}$ are the partial pressures of Ar inside and outside the tube. The applied correction was simply





(18)                     $\dot{n}_{\text{Ar sat}} = \dot{n}_{\text{Ar 0}} - \dot{n}_{\text{Ar perm}}$ .

where $\dot{n}_{\text{Ar 0}}$ and $\dot{n}_{\text{Ar sat}}$ denote respectively the argon molar flow rates at the entrance and exit of the saturator. If the outside pressure had remained at 100 kPa while the tube's interior pressure was at 108 kPa, the loss of argon would have been continuous, with the value

(19)                     $\dot{n}_{\text{Ar perm}} = 1.2 \times 10^{-7} \text{ mol s}^{-1} = 0.16 \text{ sccm}$ .

In this limit, no humid gas would leave the saturator if the argon input was less than 0.16 sccm.

We did not know the value of $\dot{n}_{\text{Ar perm}}$ because the dewar was not tightly sealed, which caused its interior pressure to be somewhere between 100 kPa and 108 kPa. We therefore allowed $\dot{n}_{\text{Ar perm}}$ to be a free parameter with a value somewhere in the range $0 < \dot{n}_{\text{Ar perm}} < 0.16$ sccm. Consistent with that range, the fitted value was 0.024 sccm.

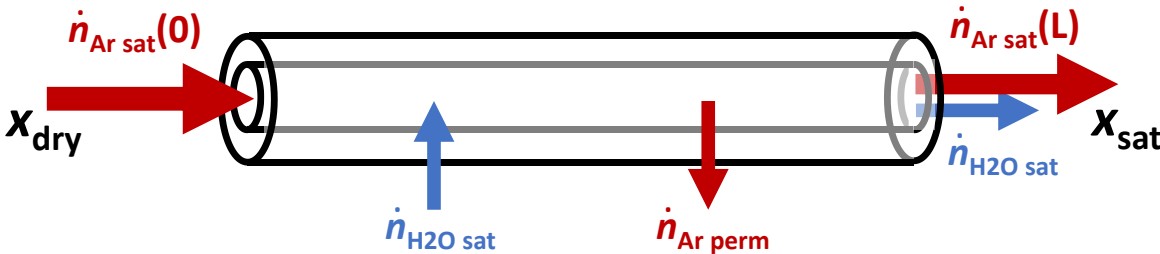


Figure 5. Some of the argon that entered the saturator was lost when it permeated the wall of the silicone tube.

The third correction accounted for the diffusion of water through argon along the saturator's exit capillary. In the absence of convective flow out of the saturator, or $\dot{n}_{\text{sat}} = 0 \text{ mol s}^{-1}$, water will diffuse from the saturator along the exit capillary at the
rate

(20)                     $\dot{n}_{\text{H2O cap}}(0) = \dfrac{DA}{L}\dfrac{P_v}{RT} = 5.1 \times 10^{-11} \text{ mol s}^{-1}$ .

Here $D$ is the diffusion coefficient of $H_2O$ vapor in Ar gas (O'Connell, 1969), $P_V$ is the vapor pressure of water at the saturator temperature, $A = 1.27 \times 10^{-6} \text{ m}^2$ is the capillary cross section area, and $L = 1.0$ m is the capillary length. The corresponding volume flow rate is

(21)                     $\dot{V}_{\text{H2O cap}}(0) = 7 \times 10^{-5} \text{ sccm}$ ,

which for a typical dry flow rate of 200 sccm corresponds to a mole fraction of 0.34 ppm.

To combine the effects of convection and diffusion, we used the steady-state one-dimensional convection-diffusion equation,

(22)                     $D\dfrac{d^2 C}{dz^2} - v\dfrac{dC}{dz} = \dfrac{dC}{dt} = 0$ ,





where $C$ is the $H_2O$ concentration in mol m$^{-3}$, $v$ is the flow velocity, and $z$ is the distance along the capillary. After setting the capillary entrance concentration at $C(0) = P_V/(RT)$ (saturation) and the exit concentration at $C(L) = 0$ (merging with the dry gas stream), one obtains the $H_2O$ flow when the convective flow is not zero,

(23)
$$\dot{n}_{H2O\ cap}(\dot{n}_{sat}) = \frac{\left(\frac{P_V}{P}\right)\dot{n}_{sat}}{1-\exp\left[-\left(\frac{P_V}{P}\right)\left(\frac{\dot{n}_{sat}}{\dot{n}_{H2O\ cap}^{(0)}}\right)\right]},$$

where the total flow rate out of the saturator is

325  (24)
$$\dot{n}_{sat} = \dot{n}_{Ar\ sat} + \dot{n}_{H2O\ sat} = \frac{\dot{n}_{Ar\ sat}}{1-\frac{P_V}{P}}.$$

This correction was significant only for mole fractions $x < 2$ ppm.

The above corrections lead to the following generalization of Eq. (11):

(25)
$$x = x_{dry} + \frac{\dot{n}_{Ar\ sat}\left(\frac{P_V}{P}\right)}{\left[\dot{n}_{dry}\left(1-\frac{P_V}{P}\right) + \dot{n}_{Ar\ sat}\right]\left\{1-\exp\left[-\left(\frac{P_V}{P}\right)\left(\frac{\dot{n}_{Ar\ sat}}{\dot{n}_{H2O\ cap}^{(0)}}\right)\right]\right\}}.$$

(The enhancement factor $f_W$, not shown here for simplicity, was included in our analysis.)

The fourth correction accounted for the pressure drops caused by flow through capillary 2 and capillary 3. These corrections were applied to the pressure $P$ in the saturator and the pressure $P_{cm}$ in the chilled-mirror hygrometer. Due to capillary 3, $P_{cm}$ was higher than the controlled pressure $P_3$ by about 1 %, or specifically

335  (26)
$$P_{cm} - P_3 = k_{cap3}\dot{n},$$

where $k_{cap3}$ is the capillary flow coefficient of capillary 3. Similarly, the pressure $P$ in the saturator was higher than $P_3$ by

(27)
$$P - P_3 = \left(k_{cap2} + k_{cap3}\right)\dot{n}.$$

As shown by **Figure 6**, the capillary flow coefficients had the similar values

(28)
$$k_{cap2} = 8.9\ \text{Pa sccm}^{-1} \quad \text{and} \quad k_{cap3} = 8.6\ \text{Pa sccm}^{-1}.$$

These values are consistent with that calculated from the length and inner radius of the capillary. We characterized the uncertainty of the capillary pressure drops by their difference. At the typical total flow rate of 200 sccm, the capillary pressure drops increased the uncertainty of the pressure $P$ in the saturator by approximately

(29)
$$u(P - P_3) = (0.3\ \text{Pa sccm}^{-1})(200\ \text{sccm}) = 60\ \text{Pa}.$$





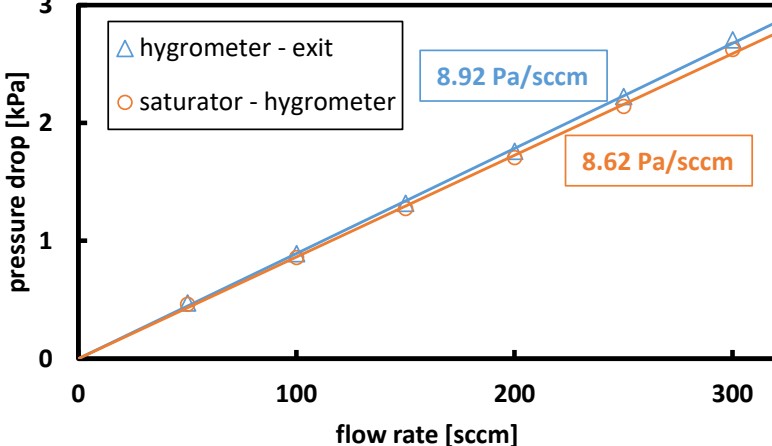

Figure 6. Similar pressure drops occurred in capillary 2 (between the saturator and the hygrometer) and capillary 3 (between the hygrometer and the exit).

## 4.4 Comparison to the chilled-mirror hygrometer

We compared the water mole fraction calculated for the generator to that measured by the chilled-mirror hygrometer. Figure 7 (upper) shows the initial comparison, which used the nominal factory calibrations for the MFCs. The points obtained with

MFC 1 and MFC 2 deviate from zero in different directions, and the points obtained with MFC 1 at different total flow rates have a scatter of ~10 %.



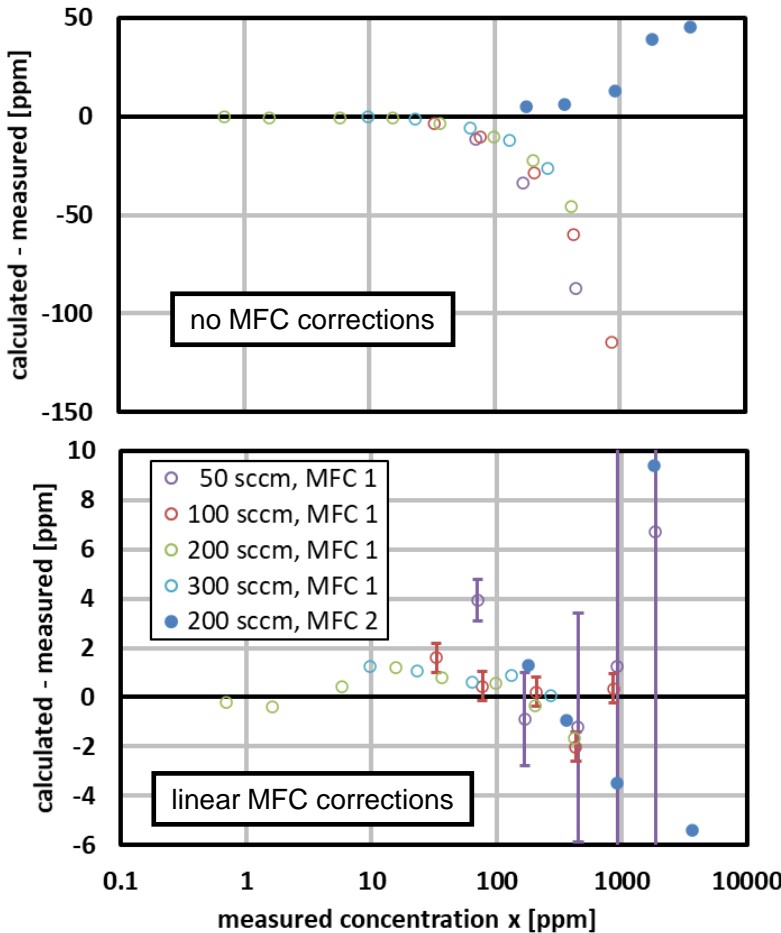

**Figure 7.** The difference of the calculated and measured mole fractions. The legend denotes the total flow rate and the MFC
that supplied gas to the saturator. UPPER: The initial comparison used the nominal factory calibrations for the MFCs. LOWER:
The final comparison used the linear MFC correction of Eq. (30). Standard uncertainty bars are displayed for two data sets at
the larger mole fractions. They correspond approximately to the manufacturer's MFC uncertainties (see Table 6); the
uncertainty of the reference hygrometer is negligible here. Note the different vertical scales.

We attributed these inconsistencies to miscalibration of the MFCs, and we attempted to correct the nominal flow rate $\dot{V}_{\mathrm{nom}}$ for
each MFC by a linear function of flow,

$$(30) \qquad \dot{V} = a + b\dot{V}_{\mathrm{nom}} \,,$$

where $a$ and $b$ are the coefficients in Table 4. The values of $b$ for MFC 1 and MFC 2, though inconsistent with the
manufacturer's specification given in Table 2, were needed to obtain agreement between similar values of humidity created
with different total flow rates. We note that the MFCs were used after the manufacturer's warranty period, and similar shifts
of the MFC calibrations were seen in the year preceding these measurements.





Table 4. MFC correction coefficients used in Eq. (30). The nominal factory calibrations corresponded to $a = 0$ and $b = 1$.

| MFC | full scale [sccm] | $a$ [sccm] | $b$ |
|---|---|---|---|
| 1  wet | 2 | 0 | 1.087 |
| 2  wet | 200 | 0 | 0.952 |
| 0  dry | 2000 | -6.0 | 1.000 |

Figure 7 (lower) shows the corrected differences, which used the MFC correction coefficients $a_0$, $b_1$, and $b_2$ shown in Table 4. This led to much better agreement, and with two exceptions (at 70 ppm and 910 ppm), the differences are within ±2 ppm for $x < 1000$ ppm. The only other adjustable parameter was the Ar permeation parameter $\dot{n}_{Ar\ perm}$. As discussed in the previous section, its fitted value fell within the range expected from the permeability of Ar through PDMS.

Figure 8 shows the calculated/measured ratio of mole fractions for the same data. With one exception (at 70 ppm), the corrected ratios fall within $1.00 ± 0.01$ for $x > 50$ ppm. At smaller mole fractions, the deviations are larger, but still correspond to a mole fraction error of approximately ±1 ppm, or a flow error of only about 1 % of the full-scale flow of MFC 1. We speculate that those deviations were caused by irreproducibility or nonlinearity not described by the linear correction of Eq. (30).

A concern is that the MFC corrections could hide an error in the model of the generator. However, there are several reasons that the corrections can be attributed to errors of the MFCs and not to an error in the model:

- The nonzero value of the offset $a_0$ (MFC 0) was necessary to obtain agreement between different total flow rates at the same mole fraction.
- The slope corrections $b_1$ and $b_2$ (MFCs 1 and 2) deviated from 1 in opposite directions; an unmodeled physical effect
likely would have caused both values to deviate in the same direction.
- The calculated/measured ratios obtained with MFC 1 for $x > 75$ ppm had a standard deviation of 0.4 %. The slope correction that minimized that scatter was $b_1 = 1.082$. The value of $b_1$ in Table 4, which was chosen for overall agreement, differs by only 0.5 %. Thus, although the slope corrections were chosen for good agreement between the generator and the reference hygrometer, they also were consistent with a measure independent of that comparison.




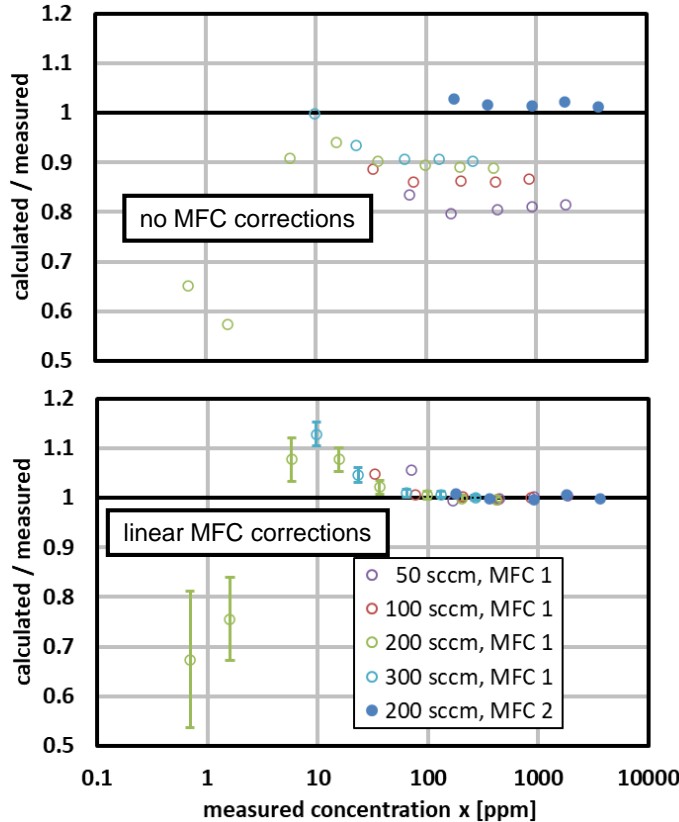

Figure 8. The ratio of the calculated and measured mole fractions obtained at total flow rates from 50 sccm to 300 sccm. UPPER: The initial comparison used the nominal calibrations for the MFCs. LOWER: The final comparison used the linear MFC correction of Eq. (30). Standard uncertainty bars are displayed for the two data sets at the smallest mole fractions. They correspond approximately to the manufacturer's MFC uncertainties (see Table 6); the uncertainty of the reference hygrometer is negligible here.

## 5 Uncertainty

The relative uncertainty of the generator is the quadrature sum of two terms,

(31)
$$\left(\frac{u(x)}{x}\right)^2 = u^2(T, P) + u^2(\text{flow}) \ ,$$

where the first term comprises the uncertainties due to the measured quantities $T$ and $P$ and the property functions $P_V(T)$ and $f_W(T,P)$,

(32)
$$u^2(T, P) = \left(\frac{T}{P_V}\frac{dP_V}{dT}\right)^2\left(\frac{u(T)}{T}\right)^2 + \left(\frac{u(P)}{P}\right)^2 + \left(\frac{u(f_W)}{f_W}\right)^2 + \left(\frac{u(P_V)}{P_V}\right)^2 \ .$$

Table 5 shows that $u(T,P)$ is only 0.17 %.





Table 5. Quantities $X$ that contribute to the uncertainty term $u^2(T,P)$ of Eq. (32). The derivative $dx/dX$ is the sensitivity of the water mole fraction $x$ to the quantity $X$.

|  | $X$ | value | $u(X)$ |  | $(dx/dX)\,u(X)$ |
|---|---|---|---|---|---|
| temperature | $T$ | 298.15 | 0.021 | K | +0.0013 |
| pressure | $P$ | 100000 | 81 | Pa | −0.0008 |
| vapor pressure | $P_V$ | 3169.8 | $0.00025\,P_V$ | Pa | +0.0003 |
| enhancement | $f_W$ | 1.0038 | 0.0006 |  | +0.0006 |
| **quadrature sum** | **$u(T,P)$** |  |  |  | **0.0017** |

The second term of Eq. (31) is the relative uncertainty of the flow ratio $\dot{n}_0/\dot{n}_{dry}$ ,

$$(33) \qquad u^2(\text{flow}) = \left(\frac{u(\dot{n}_0/\dot{n}_{dry})}{\dot{n}_0/\dot{n}_{dry}}\right)^2 .$$

If the uncertainties of the two flow meters are not correlated, then the relative uncertainties of the two MFCs add in quadrature:

$$(34) \qquad u^2(\text{flow}) = \left(\frac{u(\dot{n}_0)}{\dot{n}_0}\right)^2 + \left(\frac{u(\dot{n}_{dry})}{\dot{n}_{dry}}\right)^2 .$$

Table 6 gives values of $u(\text{flow})$ calculated for various values of $\dot{n}_0$ by using the flow uncertainties that were specified by the MFC manufacturer (Table 2). Although the present flow rates had been corrected by Eq. (30), we used the manufacturer's specifications as an estimate of the irreproducibility and nonlinearity not described by Eq. (30). In all cases $u(\text{flow}) \gg u(T,P)$, so reducing $u(\text{flow})$ would be necessary to improve the mole fraction uncertainty $u(x)$.

Table 6. Values of the relative standard uncertainty $u(\text{flow})$ of Eq. (34) calculated for various values of the saturator input flow $\dot{n}_0$ (in sccm). The uncertainties of $\dot{n}_0$ and $\dot{n}_{dry}$ are from Table 2.

| $\dot{n}_0$ full scale | $\dot{n}_0$ | $\dot{n}_{dry}$ | $10^6\,x$ | $\dfrac{u(\dot{n}_0)}{\dot{n}_0}$ | $\dfrac{u(\dot{n}_{dry})}{\dot{n}_{dry}}$ | $u(\text{flow})$ |
|---|---|---|---|---|---|---|
| 2 | 0.02 | 200 | 3 | 0.104 | 0.015 | 0.105 |
| 2 | 0.2 | 200 | 32 | 0.014 | 0.015 | 0.021 |
| 2 | 2 | 200 | 314 | 0.005 | 0.015 | 0.016 |
| 200 | 2 | 200 | 314 | 0.104 | 0.015 | 0.105 |
| 200 | 20 | 200 | 2882 | 0.014 | 0.015 | 0.021 |

While the main purpose of this article is to demonstrate the effectiveness of the silicone-tube saturator, we also note the following methods to reduce $u(\text{flow})$:

1. Use more than two MFCs to cover the desired range of flow. Meyer et al. used 7 MFCs to span a wet flow rates from 10 sccm to $10^5$ sccm (Meyer, 2008).

2. Use flow controllers or meters with smaller uncertainty. Instead of using thermal MFCs, Meyer et al. (2008) reduced their uncertainty by using commercial flow meters based on a viscous impedance.



3. Add a mass flow meter that measures the total flow. The requirement that the total flow equal the sum of the dry and wet flows allows one to characterize the flow ratio $\dot{n}_0/\dot{n}_{\mathrm{dry}}$ to within the precision of the MFCs (Takahashi, 1996).
Unfortunately, this clever method works only when the MFCs for the dry and wet flows have comparable full-scale values.

4. Compare both MFCs to a stable, well characterized flow impedance that has a pressure dependence based on theory, not calibration. Two such impedances are capillary flow meters (Berg, 2005) and critical flow nozzles (Wright, 1998).

5. Use a humidity sensor to compare the MFCs to each other by producing the same flow ratio with different flow rates.
The hygrometer would not need to be accurate, only reproducible. The MFC parameters in Table 4 were obtained partly by requiring such consistency when comparing the humidity generator with the chilled-mirror hygrometer.

## 6 Conclusions

We constructed and validated a two-flow humidity generator intended to supply a flow of humid gas at mole fractions in the range $1\ \mathrm{ppm} < x < 10^4\ \mathrm{ppm}$. It incorporates a novel saturator that comprises 5 m of silicone tubing immersed in 1 litre of water.
This simple, compact, low-cost device has proved capable of operating for months without intervention.

Characterization measurements performed against a calibrated chilled-mirror hygrometer showed that, in agreement with the known permeability of silicone, the silicone tube achieved full saturation within its operating range. They also showed that the device could serve as a primary humidity generator whose uncertainty was limited by the uncertainties of the mass flow
controllers that determined the mixing ratio. We proposed several methods to reduce that uncertainty.

## Appendix A: Permeability of H$_2$O and Ar through PDMS

### A.1 H$_2$O permeability

The permeability $\varphi$ describes the rate at which a given gas diffuses through a given solid. If a gas at pressure $p_{\mathrm{out}}$ surrounds a tube of permeability $\varphi$, length $L$, and outer and inner diameters $d_{\mathrm{out}}$ and $d_{\mathrm{in}}$ contains the same gas at pressure $p_{\mathrm{in}}$, the gas will
permeate from the outside to the inside at the molar flow rate (Crank, 1975)

$$(A1) \qquad \dot{n}_{\mathrm{perm}} = \frac{2\pi L (p_{\mathrm{out}} - p_{\mathrm{in}})}{\ln(d_{\mathrm{out}}/d_{\mathrm{in}})}\,\varphi\;.$$

Published reports of permeability measurements can be confusing when non-SI units are used. A common unit for permeability is the "barrer", which has the units

$$(A2) \qquad 1\ \mathrm{barrer} = \frac{(\mathrm{cm}^3_{\mathrm{STP}}\ \mathrm{s}^{-1}\ \mathrm{at\ STP})}{\mathrm{cm\ (cm\ Hg)}} = \frac{(\mathrm{cm}^3_{\mathrm{STP}}\ \mathrm{s}^{-1}\ \mathrm{at\ STP})}{\mathrm{cm\ (Torr/10)}} = 3.1 \times 10^{-6}\,\frac{\mathrm{mol\ s}^{-1}}{\mathrm{m\ Pa}}\;.$$





where 1 $cm^3_{STP}$ is the number of moles of an ideal gas contained in one cubic centimeter at standard temperature and pressure, often 1 atmosphere and 25 °C. Although the articles by Hamilton (1967) and Robb (1968) use this awkward unit, they are useful because they include credible descriptions of equipment and procedures. Table A1 gives values of $\varphi$ for various polymer materials.

Table A1. Published SI values of permeabilities at 25 °C for water through high density polyethylene (HDPE), polyfluoroalkoxy alkane (PFA), polypropylene (PP), low density polyethylene (LDPE), and polydimethylsiloxane (PDMS). The value from Section 4.1 has an uncertainty of $50 \times 10^{-14}$ mol s$^{-1}$ m$^{-1}$ Pa$^{-1}$.

| source | material | $\varphi$ ($10^{-14}$ mol s$^{-1}$ m$^{-1}$ Pa$^{-1}$) |
|---|---|---|
| 1967 Hamilton (Fig 8) | HDPE | 0.6 |
| 2017 McKeen (Table 11.36) | PFA | 0.6 |
| 1967 Hamilton (Fig 8) | PP | 0.8 |
| 1967 Hamilton (Fig 8) | LDPE | 1.8 |
| 1968 Robb (Table 2) | PDMS | 1100. |
| 2009 Lee (Fig 3) | PDMS | 204. |
| 2011 Velderrain (Table 3) | PDMS | 1180. |
| 2021 this work | PDMS | 950. |

The permeability of PDMS (silicone) is largest by a factor of 1000, so it is best for achieving the limit $L \gg L_{sat}$.


**A.2 Carrier gas permeability**

If the carrier gas is argon, it will diffuse out of the PDMS tube and into the surrounding water at the rate

(A3)  $$\dot{n}_{perm,Ar} = \frac{2\pi L (p_{out} - p_{in})}{\ln(d_{out}/d_{in})} \varphi_{Ar} ,$$

where $\varphi_{Ar}$ is the permeability of argon through PDMS (Robb, 1968). In the present study, with $p_{out} - p_{in} \leq 8$ kPa, this reverse

permeation effect corresponded to a steady loss of the carrier gas of as much as 0.16 cm$^3$ min$^{-1}$. Table gives permeability values for the main air gases.

Table A2. Permeability $\varphi$ and diffusivity $D$ of gases in PDMS at 25 °C (Robb1968).

| | $\varphi$ [$10^{-14}$ mol s$^{-1}$ m$^{-1}$ Pa$^{-1}$] | $D$ [$10^{-10}$ m$^2$ s$^{-1}$] |
|---|---|---|
| $N_2$ | 9 | 15 |
| $O_2$ | 19 | 16 |
| Ar | 19 | 14 |
| $H_2O$ | 1116 | |

The approach to saturation within the water reservoir will cause this effect to be time dependent, even if the reservoir is tightly sealed. At saturation, 1 L of water at 25 °C and 108 kPa can hold





(A4) $$n_{\text{perm,Ar}} = \frac{x_{\text{Ar}} m_{\text{H2O}}}{M_{\text{H2O}}} = \frac{(3.3 \times 10^{-5})(1.0 \text{ kg})}{(0.018 \text{ kg mol}^{-1})} = 1.8 \times 10^{-3} \text{ mol} ,$$

which corresponds to 41 cm$^3$ of gas at STP. (Here, $x_{\text{Ar}}$ is the solubility of Ar (O'Connell, 1969), $m_{\text{H2O}}$ is the mass of the water, and $M_{\text{H2O}}$ is the molar mass of H$_2$O.) In the present study, the time to reach saturation was estimated as

(A5) $$t_{\text{sat}} = \frac{n_{\text{perm,Ar}}}{\dot{n}_{\text{perm,Ar}}} = \frac{41 \text{ cm}^3}{0.16 \text{ cm}^3 \text{ min}^{-1}} = 4.3 \text{ h} .$$

This estimate ignores the slow diffusivity of the gas within the water, which for Ar, O$_2$ and N$_2$ is about $2 \times 10^{-9}$ m$^2$ s$^{-1}$ at 25 °C (O'Connell, 1969). The loops of silicone tubing were typically separated by 2 cm, and the associated diffusion time was estimated as

(A6) $$t_{\text{dif}} \approx \frac{l^2}{D} \approx \frac{(0.02 \text{ m})^2}{(2 \times 10^{-9} \text{ m}^2 \text{ s}^{-1})} = 56 \text{ h} .$$

**Author contributions**

Berg created the model and assisted with the apparatus construction. Chiodo constructed the apparatus, wrote the control software, and took the data. Georgin calibrated the reference hygrometer and contributed the idea of a polymer tube immersed in water.

**Competing interests**

The authors declare that they have no conflict of interest.

**Acknowledgments**

The humidity generator was created at LNE-CNAM within a research program for developing a microwave hygrometer with LNE-CETIAT. We thank Laurent Pitre for helpful discussions and for making the arrangements that allowed one of us (Berg) to work at LNE. We thank Chris Meyer for numerous constructive comments, and we thank Fernando Sparasci for obtaining

funding for the project. This project 20IND06 PROMETH2O received funding from the EMPIR programme co-financed by the Participating States and from the European Union's Horizon 2020 research and innovation programme. This work was developed within the frame of the European Metrology Research Program (EMRP) joint research project ENV07 and ENV58 "METEOMET". The EMRP is jointly funded by the EMRP participating countries within EURAMET and the European Union.





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
