# Peer review of "Silicone tube humidity generator"

_Atmospheric Measurement Techniques, 2021_

## Referee Comment (RC1)

**Content**

This manuscript describes a two-flow humidity generator based on a silicone tube humidifier. The basic principle, the design and realization of this generator is described in detail. Its ability as primary calibration standard for water vapor is shown by comparing with a reference frost point hygrometer. In addition, a comprehensive error analysis including impact of various effect is examined in detail.

**Overall impression and rating**

The overall impression of the manuscript is good. The manuscript is written in a balanced way and all aspects important for an instrument manuscript are considered. The presentation of the manuscript is excellent and nicely to read. It is well organized and the analysis and results are clearly structured and communicated in a very detailed way. However, the authors do not really show groundbreaking new results and findings, as similar techniques for humidity generators are already used and published by others. For example Hubert et al. 2016 used also polymeric material to saturate air and developed a cheep and small humidity generator. Nevertheless, I think this manuscript is still a good contribution to the scientific community as it summarize comprehensively all the effets important for a precise humidity generation. For these reasons, I recommend publication in AMT.

I have only very few comments/questions which should be considered before preparing the final/revised version.

**Specific comments/questions:**

- Page 5, lines 120-123, I do not agree with the statement that the uncertainties of the individual flow meters will cancel out by using just the ratio of both flows in the mixing ratio formular. An uncertainty estimate is always positive and negativ and both MFC have an uncorrelated uncertainty (as you write also later in Section 5). Thus, the ratio leads on to even larger total uncertainties of slightly more than 2% by using Gaussian error propagation, if you assume an uncertainty of 2% for each flow.

- page 6, formular 8, Can you quickly explain the derivation of the right part of the formular. The left part is clear but I don't see directly were the right part comes from.

- page 7/8, lines 170-172, It is not really clear why you use capillaries for the connection within and outside of the generator. It is clear as you also wrote, that the surface of a capillary is smaller compared to a thicker tubing. But for the effect of the walls (adsorption and desorption) it should be rather the ratio between air volume and surface. The volumen of the tube scales with $1/r^2$ and the surface just with $1/r$. I would assume a tubing with a thicker diameter would be better. Can you please comment that.

- page 10, lines 225-226, Where do you get the enhancement factors for $f_w(Ar)$ and $f_w(air)$ from? Maybe you can put a reference here.

- Page 15/16, lines 349-361, Did you apply any gas correction factor (GCF), if you take the standard output of the factory calibrated MFC ? Usually, the MFC are calibrated using pure N2 and if you use another carrier gas (in your Ar) you have to apply the specific GCF. This methode is not described in the manuscript and can also account partially for the discrepancies between calculated water vapor mixing ratios and the measured ones.

**Technical comments/suggestions:**

- Page 3, lines 83-84, It would be good if you could state which formular you are using for calculation of saturation water vapor pressure $P_v$.

- Page 5, Figure 1, I suggest to add the word "dry" to the label "carrier gas" in the figure.

- Page 6, lines 145-146: The Sentence seems to be not complete after the word kelvin.#

- Page 10, line 237, The saturator temperature was rather 28.5°C instead of 28°C as you write later in the manuscript. You should directly mention the right number here.

- Page 11, Figure 4: I would recommend to add error bars to the measurement points. In addition, the reference hygrometer from wich the measured points are coming from should be mentioned in the Figure caption.

- Page 12, lines 282: I recommend to add the wording "bypass the saturator" in this sentence, because you named this flow path "bypass" in Figure 2.

- Page 13, Figure 5: Figure 5 is not mentioned in the text at all. Please add at least one sentence or a reference somewhere in Section 4.3.

- Page 15, Figure 6: Please use consistent unity brackets in all the Figures. In Figure 4 you used curved brackets and in this and Fig. 7/8 you use square brackets.

- Page 20, Section 5/6, It would be nice, if you can state your total uncertainty estimate in the text of Section 5 and the conclusion section. That would help the reader directly get this important peace of information without going deeply though your detailed uncertainty estimate in Section 5.

**References**

- Hubert, T.; Tiebe, C.; Detjens, M. & Majewski, J. On-site calibration system for trace humidity sensors Measurement, 2016, 91, 251-257

---

## Author Comment (AC1)

**Polymer tube humidity generator**
**Responses of authors to AMT referees**

9 November 2021

**Referee #1**

The overall impression of the manuscript is good. The manuscript is written in a balanced way and all aspects important for an instrument manuscript are considered. The presentation of the manuscript is excellent and nicely to read. It is well organized and the analysis and results are clearly structured and communicated in a very detailed way. However, the authors do not really show groundbreaking new results and findings, as similar techniques for humidity generators are already used and published by others. For example Hubert et al. 2016 used also polymeric material to saturate air and developed a cheep and small humidity generator.

References
• Hubert, T.; Tiebe, C.; Detjens, M. & Majewski, J. On-site calibration system for trace humidity sensors Measurement, 2016, 91, 251-257

We thank the reviewer for bringing the article by Hübert et al. to our attention. Our work certainly overlaps with theirs, and we now mention their article in the appendix as well as in the introduction. Nevertheless, our use of a silicone tube long enough to achieve saturation is important because it allowed the generator to be used as a primary standard without having to measure the permeability of the tube.

We added the following paragraph to the introduction:
Some previous humidity generators have used a polymer tube to add moisture to a gas stream. Hübert et al. (2016) described a humidity calibration system based on a polymer tube immersed in a thermostatted water reservoir. However, unlike the present generator, the gas exiting their polymer tube was not saturated, and consequently the humidity at the tube exit depended on the tube's dimensions and permeability as well as the gas flow rate. Georgin described a humidity step generator that used permeable tubing, which also did not use saturated gas (Georgin, 2019). Similarly, the humidity produced by permeation-tube generators is not saturated. Commercial examples can be found in (Permeation-tube generators); see also (McKinley 2008), (Miller, 2008), and (Scace, 2008).

Nevertheless, I think this manuscript is still a good contribution to the scientific community as it summarize comprehensively all the effets important for a precise humidity generation. For these reasons, I recommend publication in AMT.

I have only very few comments/questions which should be considered before preparing the final/revised version.

Specific comments/questions:

1) Page 5, lines 120-123, I do not agree with the statement that the uncertainties of the individual flow meters will cancel out by using just the ratio of both flows in the mixing ratio formular. An uncertainty estimate is always positive and negativ and both MFC have an uncorrelated uncertainty (as you write also later in Section 5). Thus, the ratio leads on to even larger total uncertainties of slightly more than 2% by using Gaussian error propagation, if you assume an uncertainty of 2% for each flow.

The reduced uncertainty of a flow ratio can occur when the Type B uncertainties of the two MFCs are correlated. Here is an example: A primary standard has a Type A (statistical) uncertainty of 0.1 % that is random and a Type B (systematic) uncertainty of 1 % that is constant. Two MFCs calibrated by the standard will each have an uncertainty of slightly more than 1 %. Even so, both MFCs will have nearly the same relative error, which will cancel out of the ratio. When the first MFC measures a true flow of 1 sccm it will report $1.01 \pm 0.001 \rightarrow 1.009$ sccm, and when the second MFC measures a true flow of 100 sccm it will report $101 \pm 0.1 \rightarrow 101.1$ sccm. The reported ratio will be something like
$$\frac{101.1}{1.009} \cong 100.2 \ .$$
We hope this explanation satisfies the referee. Regardless, the sentence in question was deleted due to the distraction that it can cause.

2) Page 6, formular 8, Can you quickly explain the derivation of the right part of the formular. The left part is clear but I don't see directly were the right part comes from.
By total flow out of the saturator is the sum of the input dry flow and the added water flow:
$$\dot{V}_{wet} = \dot{V}_0 + \dot{V}_{H2O} \ .$$
Write this as
$$\dot{V}_{wet} = \dot{V}_0 + \left(\frac{\dot{V}_{H2O}}{\dot{V}_{wet}}\right) \dot{V}_{wet} \ .$$
For an ideal gas, the ratio of molar flow rates is the ratio of partial pressures, so that
$$\dot{V}_{wet} = \dot{V}_0 + \left(\frac{p_{wet}}{P}\right) \dot{V}_{wet} \ .$$
Solving for $\dot{V}_{wet}$ gives
$$\dot{V}_{wet} = \frac{\dot{V}_0}{1 - p_{wet}/P} \ .$$

The following sentence was added after Eq. (8):
(Equation **Error! Reference source not found.**) comes from $\dot{V}_{H2O}/\dot{V}_{wet} = p_{wet}/P$.)

3) page 7/8, lines 170-172, It is not really clear why you use capillaries for the connection within and outside of the generator. It is clear as you also wrote, that the surface of a capillary is smaller compared to a thicker tubing. But for the effect of the walls (adsorption and desorption) it should be rather the ratio between air volume and surface. The volumen of the tube scales with 1/r^2 and the surface just with 1/r. I would assume a tubing with a thicker diameter would be better. Can you please comment that.
Consider dry gas (mole fraction $x = 0$) that enters a capillary. It will exit with a water mole fraction of approximately

$$x = \frac{\dot{n}'_{wall} A}{\dot{n}_{gas}},$$

Where $\dot{n}'_{wall}$ is the rate of water desorption per unit wall area [mol s$^{-1}$ m$^{-2}$], $A$ is the wall area, and $\dot{n}_{gas}$ is the molar flow rate of the gas [mol s$^{-1}$]. The important quantity is not the volume of the tube but the ratio of the wall area to the molar flow rate.

4) page 10, lines 225-226, Where do you get the enhancement factors for fw(Ar) and fw(air) from? Maybe you can put a reference here.

We added this sentence to Section 3.5:

As mentioned in Section 2.3, we used literature values for $f_w$(air), and we derived the values of $f_w$(Ar) from values of the argon-water virial coefficient $B_{aw}$.

5) Page 15/16, lines 349-361, Did you apply any gas correction factor (GCF), if you take the standard output of the factory calibrated MFC ? Usually, the MFC are calibrated using pure N2 and if you use another carrier gas (in your Ar) you have to apply the specific GCF. This methode is not described in the manuscript and can also account partially for the discrepancies between calculated water vapor mixing ratios and the measured ones.

A good question, but the MFCs were calibrated for argon. We added this sentence to Section 3.3:

All three MFCs were calibrated at the factory for argon flow.

Technical comments/suggestions:
6) Page 3, lines 83-84, It would be good if you could state which formular you are using for calculation of saturation water vapor pressure Pv.

Done. In section 2.1 we added a citation to (Wagner, 2002).

7) Page 5, Figure 1, I suggest to add the word "dry" to the label "carrier gas" in the figure.

Done.

8) Page 6, lines 145-146: The Sentence seems to be not complete after the word kelvin.#

The sentence now reads:

… international project to redefine the kelvin in terms of the Boltzmann constant (Pitre, 2017).

9) Page 10, line 237, The saturator temperature was rather 28.5°C instead of 28°C as you write later in the manuscript. You should directly mention the right number here.

Done.

10) Page 11, Figure 4: I would recommend to add error bars to the measurement points. In addition, the reference hygrometer from wich the measured points are coming from should be mentioned in the Figure caption.

The first sentence of the caption now includes "measured by the reference hygrometer".

We added also:

The measurement uncertainties are comparable to the size of the data points.

11) Page 12, lines 282: I recommend to add the wording "bypass the saturator" in this sentence, because you named this flow path "bypass" in Figure 2.

Done.

12) Page 13, Figure 5: Figure 5 is not mentioned in the text at all. Please add at least one sentence or a reference somewhere in Section 4.3.

Done.

13) Page 15, Figure 6: Please use consistent unity brackets in all the Figures. In Figure 4 you used curved brackets and in this and Fig. 7/8 you use square brackets.

Done.

14) Page 20, Section 5/6, It would be nice, if you can state your total uncertainty estimate in the text of Section 5 and the conclusion section. That would help the reader directly get this important peace of information without going deeply though your detailed uncertainty estimate in Section 5.

This paragraph was added to the Conclusions:
An uncertainty analysis based on the MFC specifications found that generator's uncertainty varied approximately from 10 % to 2 % in the mole fraction range 3 ppm $< x <$ 3000 ppm. However, the MFCs had errors that exceeded their specifications, likely because they were used outside the manufacturer's warranty period. We corrected those errors by a linear function of flow rate. At mole fractions above 50 ppm the generator's output mole fraction $x_{gen}$ agreed to within 1 % with the value $x_{cm}$ measured by a calibrated chilled-mirror hygrometer; in other words, their ratio fell in the range $x_{gen}/x_{cm} = 1.00 \pm 0.01$. At mole fractions below 50 ppm, their differences fell in range $x_{gen} - x_{cm} = \pm 1$ ppm.

**Referee #2**

General Comments;

Overall, this is a useful piece of work, which makes humidity generation accessible with less capital investment. The uncertainty of the resulting reference gases are possibly sufficient for applications such as atmospheric research, but may struggle to determine acceptance criteria for tighter specifications.

Specific Comments;

Line 26 states that "A hygrometer can be calibrated by comparing it to one that either is a primary standard..". In my view, a hygrometer is never a primary standard, as it relies on reference humidity values from a generator for its traceability.

A device can be a primary standard if it relies on the value of a known physical quantity. For example, a rigid container can be a primary standard for volume if its volume is determined by filling it with water and weighing it. The known physical quantity is the density of water. Similarly, a chilled-mirror hygrometer is a primary standard because it is not calibrated and instead relies on the known vapor pressure curve of water.

Page 2, lines 30 to 44 provide an overview of other humidity generation methods. Typical uncertainties and ranges should be included for completeness.

Done. We added sentences that mention the performances of the generators by Cuccaro et al. (2018) and by Meyer et al. (2008).

Lines 122 to 124: I'm not entirely clear on this approach, as the MFCs appear to have been calibrated over different ranges. Unless the same standard is used over the full range, the error is linear throughout and stability and repeatability are not an issue, it would be difficult to cancel out any errors.

The phrase about the uncertainty of a flow ratio was deleted. However, please see our response above to comment (1) by Referee #1.